# Taxonomic Characterization and Secondary Metabolite Analysis of NEAU-wh3-1: An *Embleya* Strain with Antitumor and Antibacterial Activity

**DOI:** 10.3390/microorganisms8030441

**Published:** 2020-03-20

**Authors:** Han Wang, Tianyu Sun, Wenshuai Song, Xiaowei Guo, Peng Cao, Xi Xu, Yue Shen, Junwei Zhao

**Affiliations:** 1Key Laboratory of Agricultural Microbiology of Heilongjiang Province, Northeast Agricultural University, No. 600 Changjiang Road, Xiangfang District, Harbin 150030, China; wanghan507555536@gmail.com (H.W.); sty1561214024@163.com (T.S.); wenshuaisong@163.com (W.S.); guoweizi@hotmail.com (X.G.); cp511@126.com (P.C.); xuxi1758899581@126.com (X.X.); 2College of Science, Northeast Agricultural University, No. 600 Changjiang Road, Xiangfang District, Harbin 150030, China

**Keywords:** Rhizosphere soil, *Embleya*, NEAU-wh-3-1, compound, antitumor activity, antibacterial activity

## Abstract

Cancer is a serious threat to human health. With the increasing resistance to known drugs, it is still urgent to find new drugs or pro-drugs with anti-tumor effects. Natural products produced by microorganisms have played an important role in the history of drug discovery, particularly in the anticancer and anti-infective areas. The plant rhizosphere ecosystem is a rich resource for the discovery of actinomycetes with potential applications in pharmaceutical science, especially *Streptomyces*. We screened *Streptomyces*-like strains from the rhizosphere soil of wheat (*Triticum aestivum* L.) in Hebei province, China, and thirty-nine strains were obtained. Among them, the extracts of 14 isolates inhibited the growth of colon tumor cell line HCT-116. Strain NEAU-wh-3-1 exhibited better inhibitory activity, and its active ingredients were further studied. Then, 16S rRNA gene sequence similarity studies showed that strain NEAU-wh3-1 with high sequence similarities to *Embleya scabrispora* DSM 41855^T^ (99.65%), *Embleya hyalina* MB891-A1^T^ (99.45%), and *Streptomyces lasii* 5H-CA11^T^ (98.62%). Moreover, multilocus sequence analysis based on the five other house-keeping genes (*atp*D, *gyr*B, *rpo*B, *rec*A, and *trp*B) and polyphasic taxonomic approach comprising chemotaxonomic, phylogenetic, morphological, and physiological characterization indicated that the isolate should be assigned to the genus *Embleya* and was different from its closely related strains, therefore, it is proposed that strain NEAU-wh3-1 may be classified as representatives of a novel species of the genus *Embleya*. Furthermore, active substances in the fermentation broth of strain NEAU-wh-3-1 were isolated by bioassay-guided analysis and identified by nuclear magnetic resonance (NMR) and mass spectrometry (MS) analyses. Consequently, one new Zincophorin analogue together with seven known compounds was detected. The new compound showed highest antitumor activity against three human cell lines with the 50% inhibition (IC_50_) values of 8.8–11.6 μg/mL and good antibacterial activity against four pathogenic bacteria, the other known compounds also exhibit certain biological activity.

## 1. Introduction

Tumor, especially malignant tumor, has become one of the major diseases, which is a serious threat to the health of people around the world [1,2]. According to records of the World Health Organization (WHO) in 2018, more than 9 million people died of cancer, which was the second leading cause of death worldwide [3]. This figure will further rise because of aging, intensification of industrialization and urbanization, lifestyle modifications, etc. [2]. Thus, the burden of cancer cannot be ignored and the search for effective anticancer drugs is urgent [4]. On the other hand, the severe cancer incidence is also an invisible spur to the development of anti-tumor drugs throughout the world. So far, chemotherapy is still an important method for cancer treatment. Among the chemotherapeutics used, antitumor antibiotics derived from natural products account for a large proportion [5,6,7]. Natural product antibiotics were derived from various source materials including terrestrial plants, terrestrial microorganisms, marine organisms, and some invertebrates [8].

Microbial natural products have, in fact, been an excellent resource for drug discovery, particularly in the anticancer and anti-infective areas [9,10,11]. The phylum Actinobacteria accounts for a high proportion of soil microbial biomass and contains the most economically significant prokaryotes, producing more than half of the bioactive compounds in a literature survey, including antibiotics, antitumor agents, and enzymes [8,12,13,14]. Many famous antibiotics, such as bleomycin (BLM), mitomycin, anthracyclines [15], actinomycin D (ActD), polyether ionophore antibiotics, tetracyclines, quinolones, and so on, are derived from actinomycetes, which played an important role in the drug market [16]. Zincophorin, also referred to as M144255 or griseochellin, is a polyoxygenated ionophoric antibiotic [17], and has been reported to possess strong activity against Gram-positive bacteria and have strong cytotoxicity against human lung carcinoma cells A549 and Madin-Darby canine kidney cells MDCK [17,18], which was also isolated from actinomycetes. As the main genus of Actinobacteria, *Streptomyces* is the largest antibiotic producer. More than 70% of nearly 10,000 microbial origin compounds are produced by *Streptomyces* while some rare actinobacterial genera only accounted for less than 30% [19,20,21,22]. As abundant resources of larger number and wider variety of new antibiotics, *Streptomyces* strains have been continuously noted rather than any other actinomycete genera [19,23]. *Streptomyces* are widely distributed in terrestrial ecosystems, especially in the soil [24,25]. However, as time goes on, the possibility of finding novel compounds from *Streptomyces* in conventional soil has decreased and the rediscovery rate is high [22,26]. In recent years, studies on actinomycetes from diverse habitats have suggested new chemical structures and bioactive compounds [27,28]. Rhizosphere soil, the thin layer of soil around the roots of plants, has been a potential region for the discovery of functional microbes due to its special ecological environment. As early as the beginning of the last century, Hiltner proposed that there are more microorganisms in rhizosphere soil than surrounding soil [29,30,31]. There is a close relationship between rhizosphere microorganisms and plants. Plants can release organic compounds and signal molecules through root secretions to recruit microbial flora that are beneficial to their own growth. Microbes can control plant pathogens and pests by synthesizing multiple antibiotics, thereby indirectly promoting plant growth [32,33,34]. In recent years, many biologically active microorganisms and active substances produced by their secondary metabolism have been isolated from plant rhizosphere soil [35,36,37,38].

The genus *Embleya*, was very recently transferred from genus *Streptomyces* and established by Nouioui et al [39] and is a new member of the family *Streptomycetaceae* in the order *Streptomycetales* [39,40]. *Embleya* forms well-branched substrate mycelia with long aerial hyphae in open spirals and contains LL-diaminopimelic acid in the cell wall peptidoglycan, MK-9(H_4_) or MK-9(H_6_) as the major isoprenoid quinone and phosphatidylethanolamine (PE) as the predominant phospholipid [41], which is very similar to that of *Streptomyces* [42]. At present, the genus comprises only two species: *Embleya scabrispora* and *Embleya hyaline*. *Embleya scabrispora* was originally proposed as *Streptomyces scabrisporus* sp. nov. [43], and it has been reclassified to the genus *Embleya* as the type species [39,40], it could produce hitachimycin with antitumor, antibacterial, and antiprotozoal activities [44,45,46]; and *Embleya hyaline* was first described as *Streptomyces hyalinum* [41,47], and it has been reported to produce nybomycin which is an effective agent against antibiotic-resistant *Staphylococcus aureus* and it was called a reverse antibiotic [48]. 

In this study, an *Embleya* strain, NEAU-wh-3-1, with better antitumor activity was isolated from the wheat rhizosphere soil. The taxonomic identity of strain NEAU-wh3-1 was determined by a combination of 16S rRNA gene sequence and five other house-keeping genes (*atp*D, *gyr*B, *rpo*B, *rec*A, and *trp*B) analysis with morphological and physiological characteristics. The active substances of strain NEAU-wh-3-1 were also isolated, identified, and determined. Furthermore, the cytotoxicity and antimicrobial activity of the isolated compounds were tested.

## 2. Materials and Methods

### 2.1. Isolation of Streptomyces-Like Strains 

Rhizosphere soil of wheat (*Triticum aestivum* L.) was collected from Langfang, Hebei Province, Central China (39°32′ N, 116°40′ E). The soil sample should be protected from light and air-dried at room temperature for 14 days before isolation for *Streptomyces*-like strains. After drying, the soil sample was ground into powder and then suspended in sterile distilled water followed by a standard serial dilution technique. The diluted soil suspension was spread on humic acid-vitamin agar (HV) [49] supplemented with cycloheximide (50 mg L^−1^) and nalidixic acid (20 mg L^−1^). After 28 days of aerobic incubation at 28 °C, colonies were transferred and purified on the International *Streptomyces* Project (ISP) medium 3 [50], and maintained as glycerol suspensions (20%, *v/v*) at −80 °C for long-period preservation.

### 2.2. Screening of Strains with Antitumor Activity 

All the isolated were cultured on ISP medium 2 (yeast extract-malt extract agar) and incubated at 28 °C for 7 days. The spores of the strains were transferred into 250 mL Erlenmeyer flasks containing 30 mL of the production broth containing maltodextrin 4%, lactose 4%, yeast extract 0.5%, and MOPS 2%, at pH 7.2–7.4. on a rotary shaker at 250 r.p.m at 28 °C. After seven days, the production broth was extracted with an equal volume of methanol for approximately 24 h. After filtration, the filtrate substances were evaporated under reduced pressure at 50 °C to yield the crude extract and then dissolved in DMSO (dimethyl sulfoxide) at concentrations of 20 μg/mL and 100 μg/mL. The HCT-116 (human colorectal carcinoma) cell lines were maintained in Dulbecco’s modified Eagle’s medium supplemented with 10% (*w/v*) fetal bovine serum in a humidified incubator at 37 °C of 5% CO_2_ incubator. The antitumor activities of extracts with two concentrations were investigated by the SRB (Sulforhodamine B) colorimetric method. Briefly, treated cells were harvested and seeded at a density of 5 × 10^4^ cells/well into a sterile flat bottom 96-well plate for 24 h, the cells were treated with different concentrations of the extracts for 48 h and growth inhibition was measured by determining the optical density at 510 nm, and the assay was performed basing on an established method [51].

### 2.3. Morphological and Physiological and Biochemical Characteristics of NEAU-wh3-1

Gram staining was carried out by using the standard method and morphological characteristics were observed by light microscopy (Nikon ECLIPSE E200, Nikon Corporation, Tokyo, Japan) and scanning electron microscopy (Hitachi SU8010, Hitachi Co., Tokyo, Japan) using cultures grown on ISP 3 agar at 28 °C for 2 weeks. Samples for scanning electron microscopy were prepared as described by Jin et al. [52]. Growth at different temperatures (4, 10, 15, 20, 25, 28, 32, 37, 40, and 45 °C) was determined on ISP 3 medium after incubation for 14 days. Growth tests for pH range (pH 4.0–12.0, at intervals of 1.0 pH unit) using the buffer system described by Zhao et al. [53] and tolerance of various NaCl concentrations (0–10%, with an interval of 1%, *w/v*) were tested in GY (Glucose-yeast extract powder) medium (glucose 1%, yeast extract 1%, K_2_HPO_4_ 3H_2_O 0.05%, MgSO_4_ 7H_2_O 0.05%, *w/v*, pH 7.2) at 28 °C for 14 days on a rotary shaker. Hydrolysis of Tweens (20, 40, and 80) and production of urease were tested as described by Smibert and Krieg [54]. The utilization of sole carbon and nitrogen sources, decomposition of cellulose, hydrolysis of starch and aesculin, reduction of nitrate, coagulation and peptonization of milk, liquefaction of gelatin, and production of H_2_S were examined as described previously [55,56].

### 2.4. Chemotaxonomic Analysis of NEAU-wh3-1

Biomass for chemotaxonomic characterization was prepared by growing strain NEAU-wh3-1 in ISP 2 broth in shake flasks at 28 °C for 7 days. Cells were harvested by centrifugation, washed twice with distilled water, and freeze-dried. The whole-cell sugars were analyzed according to the procedures developed by Lechevalier and Lechevalier [57]. The polar lipids were examined by two-dimensional TLC (thin layer chromatography) and identified using the method of Minnikin et al. [58]. Menaquinones were extracted from freeze-dried biomass and purified according to Collins [59]. *Streptomyces lutosisoli* DSM 42165^T^ [60] was used as the reference strain for identification of menaquinones. Extracts were analyzed by a HPLC-UV method [61] using an Agilent Extend-C18 Column (150 × 4.6 mm, i.d. 5 μm) (Agilent Corp., Santa Clara, CA, USA) at 270 nm.

### 2.5. Phylogenetic Analysis of NEAU-wh3-1

Extraction of genomic DNA, PCR amplification of the 16S rRNA gene sequence and sequencing of PCR products were carried out using a standard procedure [62]. The PCR product was purified and cloned into the vector pMD19-T (Takara Bio Inc., Dalian, China) and sequenced using an Applied Biosystems DNA sequencer (model 3730XL, Applied Biosystems Inc., Foster City, California, USA). The almost complete 16S rRNA gene sequence of strain NEAU-wh3-1, comprising 1487 bp, was obtained and compared with type strains available in the EzBioCloud server [63] and retrieved using NCBI BLAST (https://blast.ncbi.nlm.nih.gov/Blast.cgi;), and then submitted to the GenBank database. The phylogenetic tree was constructed based on the 16S rRNA gene sequences of strain NEAU-wh3-1 and related reference species. Sequences were multiply aligned in Molecular Evolutionary Genetics Analysis (MEGA) using the Clustal W algorithm and trimmed manually where necessary. Phylogenetic trees were generated with the neighbor-joining [64] and maximum-likelihood [65] algorithms using MEGA software version MEGA 7.0 [66]. The stability of the topology of the phylogenetic tree was assessed using the bootstrap method with 1000 replicates [67]. A distance matrix was generated using Kimura’s two-parameter model [68]. All positions containing gaps and missing data were eliminated from the dataset (complete deletion option). The *gyr*B gene was amplified with primers PF-1 and PR-2 [69] under the PCR program for 16S rRNA gene. PCR of the *atp*D, *rec*A, *rpo*B, and *trp*B genes were performed using primers and amplification conditions described by Guo et al. [70]. The sequence data were exported as single gene alignments or a concatenated five-gene alignment for subsequent analysis as described above. Trimmed sequences of the five housekeeping genes were concatenated head-to-tail in-frame in the order *atp*D (430 bp)-*gyr*B (354 bp)-*rec*A (431 bp)-*rpo*B (208 bp)-*trp*B (556 bp). Phylogenetic analysis was performed as described above. 

### 2.6. Production

The strain *Embleya* sp. NEAU-wh3-1 was grown on the ISP medium 2 (yeast extract-malt extract agar) and incubated for 6–7 days at 28 °C. The spores of the strain were transferred into two 1.0 L Erlenmeyer flasks containing 250 mL of the seed medium and incubated at 28 °C for 48 h on a rotary shaker at 250 r.p.m. All of the media were sterilized at 121 °C for 30 min. The seed culture (8%) was transferred into 60 flasks (1.0 L) containing 250 mL of production broth. The production broth was composed of maltodextrin 4%, lactose 4%, yeast extract 0.5%, MOPS 2%, at pH 7.2–7.4. The flasks were incubated at 28 °C for 7 days, shaken at 250 r.p.m.

### 2.7. Extraction and Isolation

The final 15.0 L production broth was filtered to separate supernatant and mycelial cake. The supernatant was subjected to a Diaion HP-20 resin column and eluted with 95% EtOH. The mycelial cake was washed with water (3 L) and subsequently extracted with MeOH (3 L) to obtain soluble material. The MeOH extract and the EtOH eluents were evaporated under reduced pressure at 50 °C to yield the crude extract (24 g). The crude extract was chromatographed on a silica gel column and eluted with a stepwise gradient of CH_2_Cl_2_/MeOH (95:5/90:10/85:15/80:20/70:30/65:35, *v/v*) and giving three fractions (Fr.1–Fr.3) based on the TLC profiles, which was performed on silica-gel plates with solvent system of CHCl_3_/MeOH (9:1, *v/v*). The Fr.1 was subjected to a Sephadex LH-20 column eluted with CH_2_Cl_2_/MeOH (1:1, *v/v*) and detected by TLC to give two subfractions (Fr.1-1-Fr.1-2). The Fr.1-1 was further isolated by semi-preparative HPLC (Agilent 1100, Zorbax SB-C18, 5 μm, 250 × 9.4 mm inner diameter; 1.5 mL min^−1^; 254 nm; Agilent, PaloAlto, CA, USA) eluting with CH_3_CN/H_2_O (90:10, *v/v*) to give compound **1** (t_R_ 25.06 min, 10.5 mg), the Fr.1-2 was further isolated by preparative HPLC (Shimadzu LC-8 A, Shimadzu-C18, 5 μm, 250 × 20 mm inner diameter; 20 mL min^−1^; 220 /254 nm; Shimadzu, Kyoto, Japan) eluting with a stepwise gradient MeOH/H_2_O (30–80%, *v/v* 30 min), and giving compound **2** (t_R_ 12.7 min, 7.5 mg), compound **3** (t_R_ 17.5 min, 12.7 mg) and compound **4** (t_R_ 22.6 min, 16.3 mg). The Fr.2 was subjected to another silica gel column eluted with n-hexane/acetone (95:5-60:40, *v/v*) and further purified by semi-preparative HPLC (Agilent 1100, Zorbax SB-C18, 5 μm, 250 × 9.4 mm inner diameter; 1.5 mLmin^−1^; 254 nm; Agilent, PaloAlto, CA, USA) eluting with CH_3_CN/H_2_O (75:25, *v/v*) to give compound **5** (t_R_ 15.1 min, 13.5 mg) and compound **6** (t_R_ 24.3 min, 18.5 mg). Fr.3 was treated by an another silica gel column and eluted with a stepwise gradient of n-hexane/acetone (100:0-40:60, *v/v*) to give three fractions Fr.3-1–Fr.3-3 according to their TLC profiles, which was observed on silica-gel plates with solvent system of n-hexane/acetone (1:3, *v/v*). The Fr.3-3 was further purified by semi-preparative HPLC (Agilent 1260, Zorbax SB-C18, 5 μm, 250 × 9.4 mm inner diameter; 1.5 mL min^−1^; 220nm; 254 nm; Agilent, PaloAlto, CA, USA) eluting with CH_3_CN/H_2_O (45:55, *v/v*) to obtain compounds **7** (t_R_ 25.1 min, 13.0 mg) and **8** (t_R_ 30.1 min, 7.5 mg).

### 2.8. General Experimental Procedures

IR spectra were recorded on a Thermo Nicolet Avatar FT-IR-750 spectrophotometer (Thermo, Tokyo, Japan) using KBr disks. Optical rotations were measured on a Perkin-Elmer 341 polarimeter (PerkinElmer, Inc. Suzhou, China). UV spectra were recorded on a Varian CARY 300 BIO spectrophotometer (Varian, Cary, NC, USA). The HR-ESI-MS and ESI-MS were taken on a Q-TOF Micro LC-MS-MS mass spectrometer (Waters Co, Milford, MA, U.S.A.). Nuclear magnetic resonance (NMR) spectra (400 MHz for ^1^H and 100 MHz for ^13^C) were measured with a Bruker DRX-400 spectrometer (Bruker, Rheinstetten, Germany). HPLC analysis was performed on a preparative HPLC (Shimadzu LC-8 A, Shimadzu-C18, 5 μm, 250 × 20 mm inner diameter; 20 mL min^−1^; 220/254 nm; Shimadzu, Kyoto, Japan) as well as a semipreparative HPLC (Agilent 1100, Zorbax SB-C18, 5 μm, 250 × 9.4 mm inner diameter; 1.5 mL/min; 220/254 nm; Agilent, Palo Alto, CA, USA). Column chromatography were consisted of silica gel (100–200 mesh, Qingdao Haiyang Chemical Group Co., Qingdao, China) as well as Sephadex LH-20 gel (GE Healthcare, Glies, UK), which were analyzed by thin-layer chromatography (TLC). TLC was performed on silica-gel plates (HSGF254, Yantai Chemical Industry Research Institute, Yantai, China) and the developed plates were observed under a UV lamp at 254 nm or by heating after spraying with sulfuric acid-ethanol, 5:95 (*v/v*). 

### 2.9. Biological Assays

The cytotoxicity of the eight compounds was assayed by cell counting kit-8 (CCK-8) colorimetric method [71] in vitro against the human leukemia cells K562, hepatocellular liver carcinoma cell line HepG2, and the human colon tumor cell line HCT-116. The cell lines were routinely in Dulbecco’s Modified Eagle’s Medium (DMEM) containing 10% calf serum at 37 °C for 4 h in a humidified atmosphere of 5% CO_2_ incubator. The adherent cells at logarithmic phase were digested by pancreatic enzymes and inoculated onto 96-well culture plate at a density of 1.0 × 10^4^ cells per/well. Test samples and control were dissolved in DMSO (dimethyl sulfoxide) and then added to the medium, incubated for 72 h. Then, the cell counting kit-8 (CCK-8, Dojindo, Kumamoto, Japan) reagent was added to the medium followed by further incubation for 3 h. Absorbance at 450 nm with a 600 nm reference was measured thereafter using a SpectraMax M5 microplate reader (Molecular Devices Inc., Sunnyvale, CA, USA). The inhibitory rate of cell proliferation was expressed as IC_50_ values and calculated by the following formula: Growth inhibition (%) = [ODcontrol-ODtreated]/ODcontrol×100

Doxorubicin was tested as a positive control, and cell solutions containing 0.5% DMSO were tested as a negative control.

The antibacterial activities of the isolated compounds were tested against Gram-positive bacteria *Staphylococcus aureus*, *Bacillus subtilis*, and *Sarcina lutea* and Gram-negative bacteria *Klebsiella pneumoniae* and *Escherichia coli* with the minimum inhibitory concentration (MIC) method recommended by the Clinical and Laboratory Standards Institute [72].

## 3. Results

### 3.1. Isolation and Screening of an Antitumor Compound Producing Strains 

Thirty-nine strains belonging to actinomycetes were isolated from the soil samples. The crude extracts of these isolates were examined for their cytotoxic activity at dilution concentrations of 100 μg/mL and 20 μg/mL. As a result of primary screening, fourteen strains showed cytotoxic activity to human colon tumor cell line HCT-116 (Figure 1). Due to the superior cytotoxic activity of strain NEAU-wh3-1, which inhibition rate was greater than 80% at both concentrations, further chemical investigations were performed on this strain.

### 3.2. Polyphasic Taxonomic Characterization of NEAU-wh3-1

Morphological observation of 2-week-old cultures of strain NEAU-wh3-1 grown on ISP 3 medium revealed that the strain has the typical characteristics of genus *Embleya* and formed well-developed, branched substrate hyphae and aerial mycelium that differentiated into spiral spore chains consisted of cylindrical spores (0.6–0.8μm × 0.9–1.3μm), the spores were rough-surfaced and non-motile (Figure 2). Strain NEAU-wh3-1 was found to grow at a temperature range of 4 to 37 °C (optimum temperature 28 °C), pH 5 to 12 (optimum pH 7), and NaCl tolerance of 0% to 3% (optimum NaCl of 1%). The physiological and biochemical properties of strain NEAU-wh3-1, *Embleya scabrispora* DSM 41855^T^, *Embleya hyalina* MB891-A1^T^, and *Streptomyces lasii* 5H-CA11^T^ are given in Table 1**.**

Chemotaxonomic analyses revealed that strain NEAU-wh3-1 contained LL-diaminopimelic acid as cell wall diamino acid. The whole-cell sugar was found to contain arabinose, glucose, and ribose. The phospholipid profile consisted of diphosphatidylglycerol (DPG), phosphatidylethanolamine (PE), phosphatidylinositol (PI), and two unidentified lipids (ULs) (Appendix A). The menaquinones detected were MK-9(H_4_) (46.5%), MK-9(H_6_) (45.8%), and MK-9(H_8_) (7.7%). 

The almost complete 16S rRNA gene sequence of strain NEAU-wh3-1 (1487 bp) was determined and deposited with the accession number MN928616 in the GenBank/EMBL (European Molecular Biology Laboratory)/DDBJ (DNA Data Bank of Japan) databases. EzBioCloud analysis suggests that strain NEAU-wh3-1 shared the highest 16S rRNA gene sequence similarities with *Embleya scabrispora* DSM 41855^T^ (99.65%), *Embleya hyalina* MB891-A1^T^ (99.45%), and *Streptomyces lasii* 5H-CA11^T^ (98.62%). Phylogenetic analysis based on the 16S rRNA gene sequences indicated that the strain formed a stable cluster with *E. scabrispora* DSM 41855^T^, *E. hyalina* MB891-A1^T^, and *S. lasii* 5H-CA11^T^ based on neighbor-joining algorithm (Figure 3) and also supported by the maximum-likelihood algorithm (Appendix A). To further clarify the affiliation of strain NEAU-wh3-1 to its closely related strains, partial sequences of housekeeping genes including *atp*D, *gyr*B, *rec*A, *rpo*B, and *trp*B were obtained. GenBank accession numbers of the sequences are displayed in Appendix A. The phylogenetic tree based on the neighbor-joining tree constructed from the concatenated sequence alignment (1979 bp) of five housekeeping genes (Figure 4) suggested that the isolate clustered with *E. scabrispora* DSM 41855^T^ and *E. hyalina* MB891-A1^T^, and also supported by the maximum-likelihood algorithm (*Streptomyces lasii* 5H-CA11^T^ lacks housekeeping genes; Appendix A). Moreover, pairwise distances calculated for NEAU-wh3-1 and the related species using concatenated sequences of *atp*D-*gyr*B-*rec*A-*rpo*B-*trp*B were well above 0.007 (Appendix A) for the related species, which was considered to be the threshold for species determination [74].

### 3.3. Structural Elucidation

The strain NEAU-wh3-1 was grown preparative scale in 15.0 L of production broth for 7 days. Bioassay-guided isolation of the active components of the strain yielded eight main bioactive compounds. Compounds **2**–**8** are known compounds, which structures were elucidated as conglobatin (**2**) [75], piericidin C1 (**3**) [76], piericidin C5 (**4**) [77], piericidin A1 (**5**) [78], piericidin A3 (**6**) [76], Mer-A 2026 A (**7**) [79], and BE-52211 D (**8**) [80] by analysis of their spectroscopic data and comparison with literature values (Figure 5, Appendix A). Compound **1** is a new zincophorin analogue (Figure 6, Appendix A) [17].

Analysis of ^1^H NMR spectrum of **1** revealed the presence of three olefinic protons at *δ*_H_ 5.52 (1H, m), 5.36 (1H, m), 5.20 (1H, d, *J* = 8.9 Hz), seven aliphatic methine protons at *δ*_H_ 4.07 (1H, m), 4.06 (1H, m), 3.77 (1H, d, *J* = 8.9 Hz), 3.72 (1H, dd, *J* = 9.6, 2.1 Hz), 3.58 (1H, d, *J* = 9.2 Hz), 3.49 (1H, m), 3.28 (1H, m), seven methylene protons at *δ*_H_ 2.18 (1H, m), 2.13 (1H, m), 1.77 (1H, m), 1.67 (2H, m), 1.40 (1H, m), 1.35 (1H, m), 1.28 (1H, m), one singlet methyl at *δ*_H_ 1.63 s, in addition to eight doublet methyl protons at *δ*_H_ 1.18 (3H, d, *J* = 7.1 Hz), 1.15 (3H, d, *J* = 7.2 Hz), 1.11 (3H, d, *J* = 7.0 Hz), 0.98 (3H, d, *J* = 6.5 Hz), 0.97 (3H, d, *J* = 6.6 Hz), 0.86 (3H, d, *J* = 6.7 Hz), 0.81 (3H, d, *J* = 6.5 Hz), 0.70 (3H, d, *J* = 6.7 Hz). The ^13^C NMR and DEPT135 spectra (Table 2) of 1 showed 31 resonances attributable to a carbonyl carbon at *δ*_C_ 175.6, one *sp*^2^ quaternary carbon at *δ*_C_ 132.3, three *sp*^2^ methines at *δ*_C_ 136.6, 134.5, 132.4. In the sp^3^-carbon region, the spectrum showed six oxygenated methines at *δ*_C_ 84.3, 83.3, 81.7, 76.1, 74.0, 69.4, six methines at *δ*_C_ 42.2, 37.1, 36.5, 36.0, 33.0, 26.1, four methylenes at *δ*_C_ 34.2, 28.6, 27.1, 24.9 and nine methyl carbons at *δ*_C_ 22.9, 22.9, 17.1, 16.7, 15.4, 12.3, 11.3, 10.8, 10.1. The ^1^H–^1^H COSY correlations (Figure 6**)** of H-2/H-3/H_2_-4/H_2_-5/H-6/H-7/H-8/H-9/H-10/H-11/H-12/H-13/H_2_-14/H_2_-15/H-16/H-17/H-18/H-19 established connectivity from H-2 atom along the chain through to C-19 atom. The correlations between H-21/H-22/H_3_-23/H_3_-24, H-12/H_3_-27, H-10/H_3_-28, H-18/H_3_-26, H-8/H_3_-29 protons in the ^1^H–^1^H COSY spectrum (Figure 6**)** indicated the five structural units of C-21–C-24, C-18–C-26, C-12–C-27, C-10-C-28, C-8-C-29. The observed HMBC (heteronuclear multiple bond correlation) correlations (Figure 6**)** from H_3_-23, H_3_-24 to C-21, C-22, from H_3_-25 to C-19, C-21, from H_3_-26 to C-17, C-18, and C-19, from H_3_-27 to C-11, C-12, and C-13, from H_3_-28 to C-9, C-10, C-11, from H_3_-29 to C-7, C-8, C-9, from H_3_-30 to C-5, C-6, C-7, from H_3_-31 to C-2 and C-3, from H-21 to C-19, H-20 to C-18, from H-19 to C-17, from H_3_-23 to C-21 established the linkage of C-2–C-22. The carbonyl group was connected with C-2 by the HMBC corrections from H-2 and H_3_-31 to C-1 (*δ*_C_ 175.6). The correlations from H-7 (*δ*_H_ 3.77 d, *J* = 8.9 Hz) to C-3 (*δ*_C_ 74.0) indicated the linkage of C-3 and C-7 through an oxygen atom to form a tetrahydropyran ring. Taking the molecular formula of C_31_H_56_O_7_ into account, four hydroxyl groups were situated at C-9, C-11, C-13, C-19, respectively, and a carboxyl group was situated at C-1. Comparison the NMR data of **1** with Zincophorin [17], a mocarboxylic acid ionophore contains one single tetrahydropyran ring, which was isolated from a strain of *Streptomyces griseus*, implied that **1** was identified to be an analogue of Zincophorin, the difference between two compounds was that the terminal ethyl group in Zincophorin was replaced by a H proton in compound **1**. On the basis of the above spectroscopic data, a gross structure of **1** was established and named Zincophorin B, and the ^1^H and ^13^C resonances in **1** were assigned (Table 2).

### 3.4. Biological Activity 

The cytotoxic activities of compounds 1–8 against K562, HCT-116, and HepG2 cancer cell lines are showed in Table 3. Eight compounds restrained proliferation of the tested cells and compound **1** showed the highest cytotoxic activity, and the average IC_50_ values were lower than 10.0 μg/mL.

The result of minimum inhibitory concentrations (MICs) showed that compound **1** showed good activities against Gram-positive bacteria *Staphylococcus aureus*, *Sarcina lutea*, and *Bacillus subtilis*, and the Gram-negative bacteria *Klebsiella pneumoniae* in vitro (Table 4). Compound **8** showed weak antibacterial activity against two Gram-positive bacterium and the minimum inhibitory concentrations (MICs) of compounds **2**–**7** were determined to be >10 mg/mL, so they had no activity against these tested pathogens.

## 4. Discussion

In this research, the results of morphological, physiological, and biochemical tests showed that strain NEAU-wh3-1 has typical characteristics of the genus *Embleya* [41]. Such as containing LL-diaminopimelic acid as the cell wall peptidoglycan, MK-9(H_4_), and MK-9(H_6_) as the major menaquinones, phosphatidylethanolamine (PE) as the predominant phospholipid and arabinose in the whole sugars. Moreover, strain NEAU-wh3-1 formed spiral spore chains and the spore surface was rough, which are consistent with *E**. scabrispora* DSM 41855^T^ and *E**. hyalina* MB891-A1^T^ [39,41]. In addition, the phylogenetic trees constructed from the 16S rRNA gene sequences and the concatenated sequences alignment (1979 bp) of five housekeeping genes all suggested that the isolate should be assigned to the genus *Embleya*.

However, some obvious differences could also be found between strain NEAU-wh3-1 and its closely related strains regarding several phenotypic and chemotaxonomic characteristics (Table 1). The isolate was able to grow at 4 °C, in contrast to its closely related strains, which were not. The composition of phospholipids and menaquinones of strain NEAU-wh3-1 was also different from its related species, *E. scabrispora* DSM 41855^T^, *E. hyalina* MB891-A1^T^ and *S. lasii* 5H-CA11^T^. Most notably, the whole-cell sugars of strain NEAU-wh3-1 was found to contain arabinose, glucose, and ribose, while *E. scabrispora* DSM 41855^T^ only contains arabinose, *E. hyalina* MB891-A1^T^ contains arabinose and glucose and *S. lasii* 5H-CA11^T^ contains glucose and ribose, which also could distinguish the strain from its closely related strains. Other phenotypic differences include the temperature and pH range of growth, patterns of carbon and nitrogen utilization, hydrolysis of cellulose, starch, and Tweens (20, 40, and 80), liquefaction of gelatin, peptonization of milk, production of H_2_S, and urease and reduction of nitrate. Therefore, it is evident from the phenotypic, genotypic, and chemotaxonomic data that strain NEAU-wh3-1 may represent a novel species of the genus *Embleya*.

The genus *Embleya*, recently transferred from genus *Streptomyces*, is a new member of the family *Streptomycetaceae* [39,40]. At present, it contains only two species: *Embleya scabrispora*, could produce hitachimycin with antitumor, antibacterial, and antiprotozoal activities [44,45,46]; and *Embleya hyaline*, could produce nybomycin which is an effective agent against antibiotic-resistant *Staphylococcus aureus* and is called a reverse antibiotic [48]. During the study of the chemical properties of the active ingredients of strain NEAU-wh3-1, eight active compounds were obtained, including one macrolide dilactone, five piericidins, one β-hydroxy acetamides, an analogue of monocarboxylic acid ionophore, which were observed to fit into at least three types based on their molecular skeletons. This shows to some extent that this strain has the ability to produce metabolites with a wide variety of different skeletal structures. Out of these compounds, Piericidins (**3**-**7**) were a class of polyene alpha-pyridone heterocyclic antibiotics, among them, Piericidin A1(**5**) was first reported [81], which was isolated from *Streptomyces mobaraensis*. Piericidins exhibit interesting biological activities, in particular antitrypanosomal [82]. In our research, compounds **3**-**7** exhibited different degrees of cytotoxicity on three types of tumor cells, but they did not show any antibacterial activity, which was consistent with previous reports [78,81]. As a Piericidins-producing strain, NEAU-wh3-1 has certain application potential in pest control. Conglobatin (**2**) is an unusual 16-membered macrocyclic diolide originally isolated from a polyether-producing strain of *Streptomyces conglobatus* ATCC 31005^T^ and was reported to be essentially devoid of antifungal, antibacterial, antitumor, and antiprotozoal activity at that time [75]. However, in recent research, FW-04-806 is identical in structure to conglobatin, and it has been reported to inhibit the growth of a human chronic myelocytic leukemia K562 cell line with an IC_50_ of 6.66 μg/mL, further study also investigated the effects of FW-04-806 on SKBR3 and MCF-7, respectively [83]. Its mechanism of action appears to be novel, via direct binding to the N-terminal domain of Hsp90 and disruption of its interaction with co-chaperone Cdc37 [84]. In our antitumor activity test, Conglobatin (**2**) showed good bioactivity against two tumor cell lines, supporting it at least partially accounted for the cytotoxic activity of the strain NEAU-wh3-1 extract. BE-52211 D (**8**) was a cytotoxic metabolite from a strain of *Streptomyces* and had been reported to have moderate cytotoxicity against human hepatocellular liver carcinoma cells HepG2, human leukemia cells K562, and human colon carcinoma cells HCT-116 with the IC_50_ values of >10 μg/mL [80], which is consistent with the result in the present study. Compound **1** structurally related to zincophorin, which was also referred to as M144255 or griseochellin and is a polyoxygenated ionophoric antibiotic isolating from *Streptomyces griseus* in 1984 [17]. It has been reported to possess strong in vivo activity against Gram-positive bacteria and have strong cytotoxicity against human lung carcinoma cells A549 and Madin-Darby canine kidney cells MDCK [18]. No biological activity has been reported against Gram-negative bacteria, yeasts, and fungi [85]. The second member in the zincophorin family named CP-78545, was found in the culture broth of *Streptomyces* sp. N731-45. The structural difference between them is that CP-78545 has an extra terminal double bond; but they have similar spectrum and potency on biological properties except for the antitumor activity (no reports) [86]. In our antitumor and antimicrobial assays, compound **1** showed the highest antitumor activity against three human cell lines and good antibacterial activity against Gram-negative bacteria. To our knowledge, this is the first report of this kind of compound with antibacterial activity against Gram-negative bacteria. This study has enriched the activity spectrum of Zincophorins.

## 5. Conclusions

Strain producing a new compound with strong antitumor activity, isolated from the rhizosphere soil of wheat (*Triticum aestivum* L.) in HeBei province, China. Morphological and chemotaxonomic features together with phylogenetic analysis suggested that strain NEAU-wh3-1 belonged to the genus *Embleya*. Cultural and biochemical characteristics combined with multilocus sequence analysis clearly revealed that strain NEAU-wh3-1 may represent a novel species of the genus *Embleya.* Moreover, eight compounds, including one new compound with higher antitumor activities against three human cell lines, were isolated from the strain.

## Figures and Tables

**Figure 1 microorganisms-08-00441-f001:**
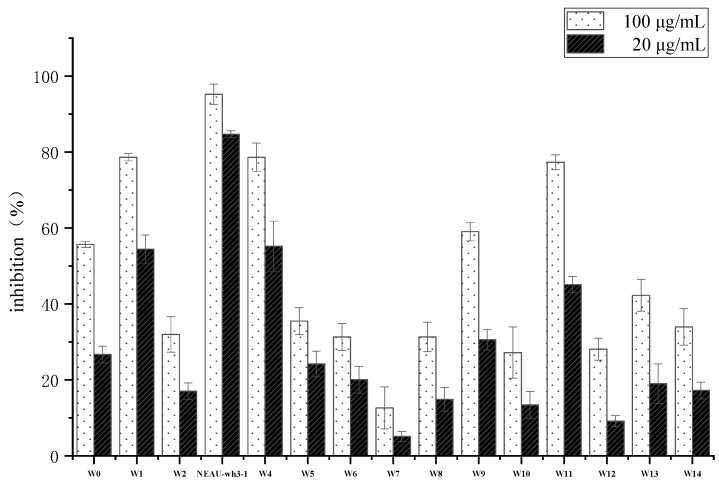
Antitumor activities of extracts obtained from fourteen isolates against human colon tumor cell line HCT-116.

**Figure 2 microorganisms-08-00441-f002:**
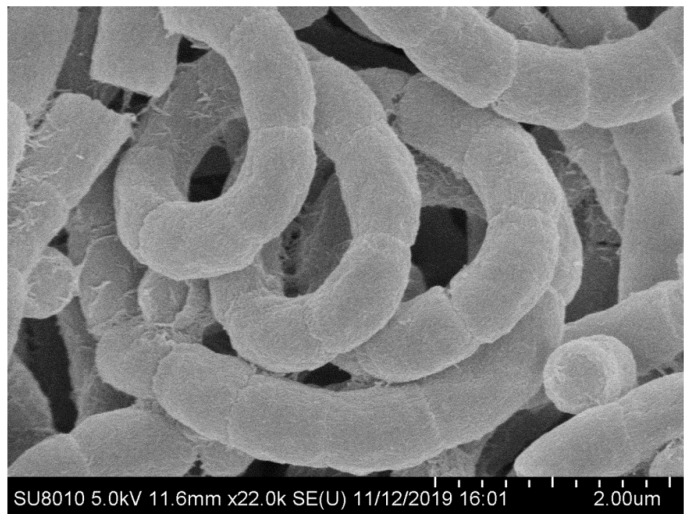
Scanning electron micrograph of spore chains of strain NEAU-wh3-1 grown on ISP 3 agar for 2 weeks at 28 °C.

**Figure 3 microorganisms-08-00441-f003:**
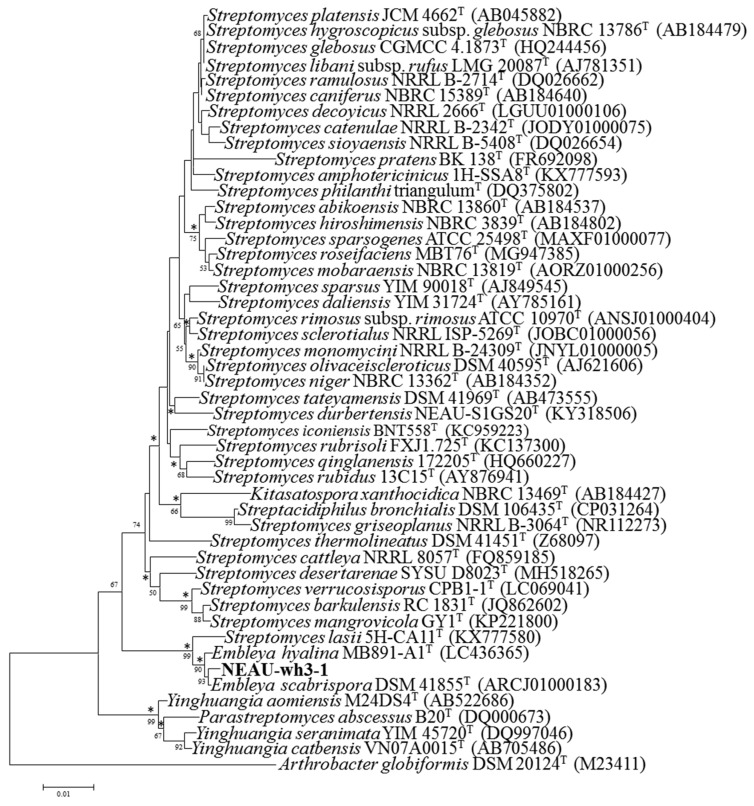
Neighbor-joining tree showing the phylogenetic position of strain NEAU-wh3-1 (1487 bp) and related taxa based on 16S rRNA gene sequences. Bootstrap values > 50% (based on 1000 replications) are shown at branch points. *Arthrobacter globiformis* DSM 20124^T^ (M23411) was used as an outgroup. *Asterisks* indicate branches also recovered in the maximum-likelihood tree; Bar, 0.01 substitutions per nucleotide position.

**Figure 4 microorganisms-08-00441-f004:**
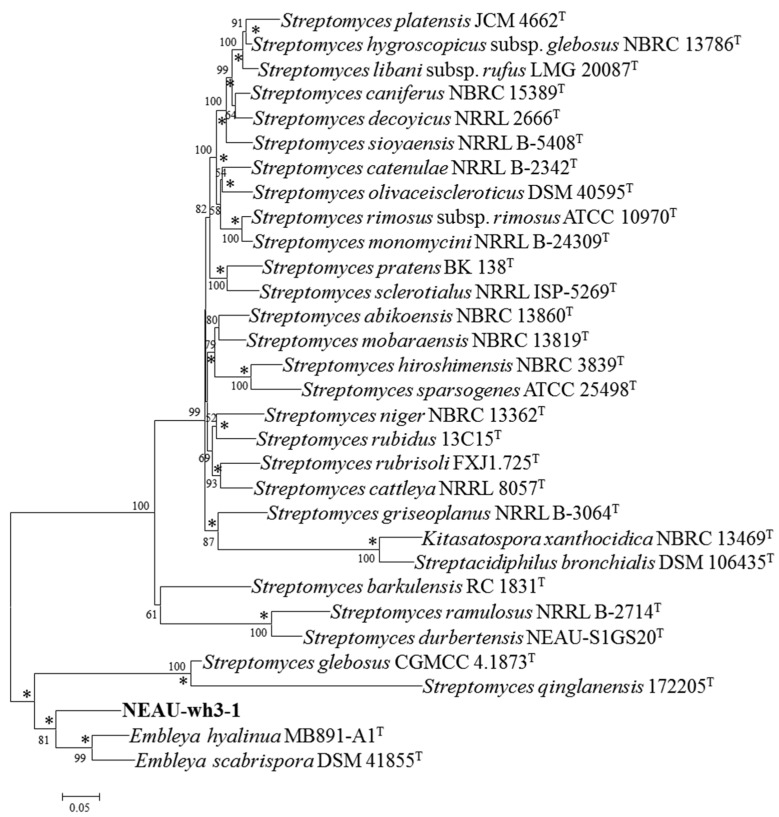
Neighbor-joining tree based on multilocus sequence analysis (MLSA) analysis of the concatenated partial sequences (1979 bp) from five housekeeping genes (*atp*D, *gyr*B, *rec*A, *rpo*B, and *trp*B) of strain NEAU-wh3-1 (in bold) with related taxa. Only bootstrap values above 50% (percentages of 1000 replications) are indicated. Asterisks indicate branches also recovered in the maximum-likelihood tree; Bar, 0.05 substitutions per nucleotide position.

**Figure 5 microorganisms-08-00441-f005:**
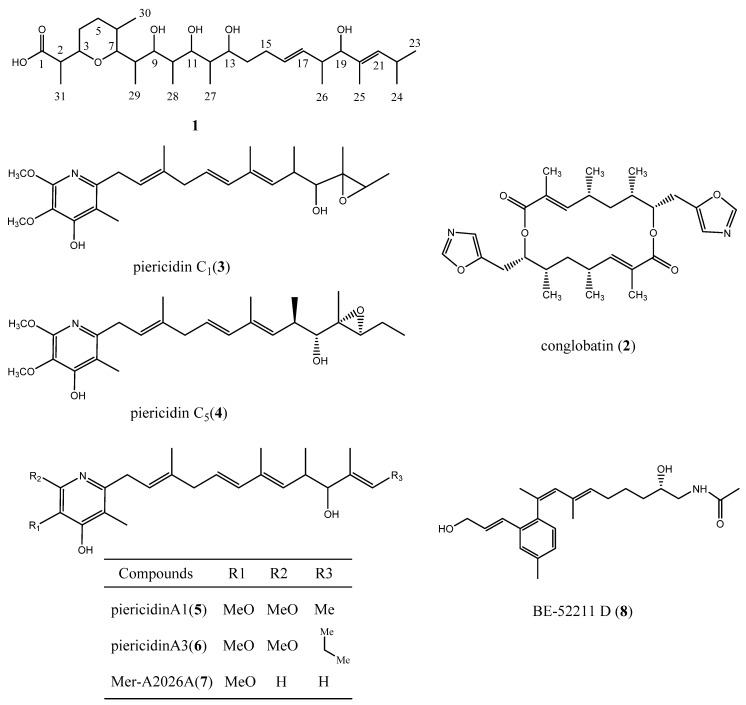
The structures of compounds **1**–**8.** Compound **1** was isolated as white solid with ${\left[\alpha \right]}_{D}^{25}$ + 15 (c 0.043, EtOH) and UV (EtOH) λmax nm (log ε): 202 (4.53). Its molecular formula was established as C_31_H_56_O_7_ by HR-ESI-MS at *m/z* 539.3942 [M-H]^-^ (calcd 539.3953 as C_31_H_55_O_7_). The IR spectrum revealed hydroxyl absorption at 3320 cm^−1^ and carbonyl absorption at 1735 cm^−1^, as well as methyl and methylene absorptions at 2953 cm^−1^ and 2924 cm^−1^.

**Figure 6 microorganisms-08-00441-f006:**
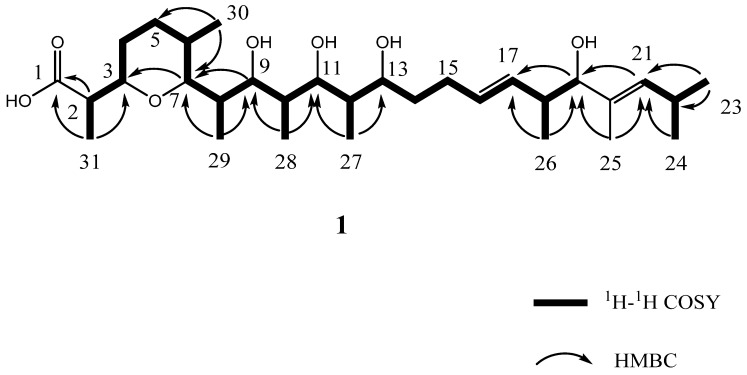
2D nuclear magnetic resonance (NMR) correlations of compound **1.**

**Table 1 microorganisms-08-00441-t001:** The physiological and biochemical properties of strain NEAU-wh3-1, *Embleya scabrispora* DSM 41855^T^, *Embleya hyalina* MB891-A1^T^, *Streptomyces lasii* 5H-CA11^T^.

Characteristic	1	2^a,c^	3^b^	4^c^
Decomposition of				
Cellulose	−	−	ND	+
Tween 20	−	+	ND	+
Tween 40	+	+	ND	+
Tween 80	−	+	ND	+
Liquefaction of gelatin	−	−	+	+
Growth temperature (°C)	4–37	18–36	10–28	15–38
pH range for growth	5–12	4–10	6–11	5–11
NaCl tolerance range (*w/v*, %)	0–3	0–3	0–1	0–2.5
Milk coagulation	−	w	−	+
Nitrate reduction	−	+	−	−
Starch hydrolysis	−	w	−	−
Carbon source utilization				
l-arabinose	+	±	−	−
Dulcitol	+	w	−	W
d-Fructose	+	−	+	−
d-Galactose	+	−	+	−
d-Glucose	+	+	+	+
Inositol	−	+	+	−
Lactose	+	+	−	+
d-Maltose	+	−	±	−
d-Mannitol	+	−	−	−
d-Mannose	+	−	+	−
d-Raffinose	+	−	−	+
d-Ribose	−	+	ND	−
d-Sorbitol	+	−	−	−
d-Sucrose	+	±	W	+
d-Xylose	+	+	−	−
l-Rhamnose	−	+	+	−
Nitrogen source utilization				
l-Alanine	+	W	ND	+
l-Arginine	+	W	ND	+
l-Asparagine	+	W	ND	−
l-Aspartic acid	+	+	ND	+
Creatine	+	w	ND	−
l-Glutamic acid	+	w	ND	+
l-Glutamine	+	w	ND	+
Glycine	+	w	ND	+
l-Proline	−	+	ND	+
l-Serine	+	−	ND	−
l-Threonine	+	w	ND	+
l-Tyrosine	+	+	ND	+
Phospholipids	DPG, PE, PI, UL	PE, PGL	DPG, PE, PI	DPG, PME, PI, PIM, GL
Menaquinones	MK-9(H_4_), MK-9(H_6_), MK-9(H_8_)	MK-9(H_2_), MK-9(H_4_), MK-9(H_6_)	MK-9(H_4_), MK-9(H_6_), MK-9(H_8_)	MK-9(H_4_), MK-9(H_6_), MK-9(H_8_)
Whole cell-wall sugars	Arabinose, glucose, ribose	Arabinose	Arabinose, glucose	Glucose, ribose

Strains: 1, NEAU-wh3-1; 2, *Embleya scabrispora* DSM 41855^T^; 3, *Embleya hyalina* MB891-A1^T^; 4, *Streptomyces lasii* 5H-CA11^T^. Abbreviation: +, positive; –, negative; ±, ambiguous; ND, not determined; w, weak; DPG, diphosphatidylglycerol; PME, phosphatidylmonomethylethanolamine; PE, phosphatidylethanolamine; PI, phosphatidylinositol; PIM, phosphatidylinositolmannoside; UL, unidentified lipid; GL, glucosamine-containing lipid; PGL, phospholipid containing glucosamine. All data are from this study except where marked. ^a^ Data from Ping et al. [43]; ^b^ Data from Komaki et al. [41]; ^c^ Data from Liu et al. [73].

**Table 2 microorganisms-08-00441-t002:** ^1^H and ^13^C NMR data of compound 1 in CDCl_3_.

No.	*δ*_H_ (*J* in Hz)	*δ*_C_ (p.p.m)	No.	*δ*_H_ (*J* in Hz)	*δ*_C_ (p.p.m)
1		175.6	16	5.52 m	132.4
2	3.28 m	37.1	17	5.36 m	134.5
3	4.07 m	74.0			
4	1.67 m	24.9	18	2.27 m	42.2
			19	3.58 d (9.2)	81.7
5a	1.28 m	27.1	20		132.3
5b	1.40 m		21	5.20 d (8.9)	136.6
6	1.52 m	26.1	22	2.59 m	26.1
7	3.77 d (8.9)	76.1	23	0.97 d (6.6)	22.9
8	2.04 m	33.0	24	0.98 d (6.5)	22.9
9	3.72 dd	84.3	25	1.63 s	10.1
	(9.6, 2.1)				
10	2.01 m	36.5	26	0.86 d (6.7)	16.7
11	3.49 m	83.3	27	1.15 d (7.2)	10.8
12	1.75 m	36.0	28	0.70 d (6.7)	12.3
			29	1.11 d (7.0)	11.3
13	4.06 m	69.4	30	0.81 d (6.5)	17.1
14a	1.35 m	34.2	31	1.18 d (7.1)	15.4
14b	1.77 m				
15a	2.13 m	28.6			
15b	2.18 m				

**Table 3 microorganisms-08-00441-t003:** The cytotoxicity of compounds 1–8.

Compound	IC_50_ (μg/mL)		
	K562	HCT-116	HepG2
1	8.8 ± 1.5	9.5 ± 0.8	9.6 ± 5.6
2 3 4 5 6 7	57.1 ± 7.3 — — 28.3 ± 1.1 36.6 ± 2.4 —	75.42 ± 2.1 68.39 ± 3.3 36.8 ± 5.6 14.3 ± 1.6 21.6 ± 4.1 112.3 ± 5.7	— 53.78 ± 6.7 17.5 ± 1.9 27.3 ± 5.8 79.7 ± 5.9 —
8	11.42 ± 3.05	15.13 ± 1.76	10.83 ± 3.47
Doxorubicin	1.1 ± 0.1	0.9 ± 0.3	2.1 ± 0.2

**Table 4 microorganisms-08-00441-t004:** The antibacterial activity of compounds 1-8.

Compounds	MIC (μg/mL)				
	Gram-Positive Bacteria			Gram-Negative Bacteria	
	*Staphylococcus aureus*	*Sarcina lutea*	*Bacillus subtilis*	*Klebsiella pneumonie*	*Escherichia coli*
**1**	31.0 ± 2.5	44.0 ± 5.8	3.5 ± 0.5	25.0 ± 1.5	—
**2–7**	—	—	—	—	—
**8**	210.0 ± 20.0	190.0 ± 15.0	—	—	—

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
