# Peer review of "Taxonomic Characterization and Secondary Metabolite Analysis of NEAU-wh3-1: An Embleya Strain with Antitumor and Antibacterial Activity"

_microorganisms, 2020, doi:10.3390/microorganisms8030441_

Round 1

Reviewer 1 Report

Intersting article.

Biological activity of compounds are not very high so for instance you can change the "strongest" of line 443 into "highest"

Author Response

Dear reviewer, Thank you for the valuable suggestions. In the following, we provide our itemized list of changes according to your suggestions and highlighted the changes in our manuscript. Thank you very much for your kindness and help. Sincerely yours, Dr. Han Wang and Prof. Junwei Zhao 1. Biological activity of compounds are not very high so for instance you can change the "strongest" of line 443 into "highest" Reply: Thank you very much and we have revised related descriptions.

Reviewer 2 Report

The manuscript describes the isolation of soil actinomycetes from the rhizosphere soil of wheat (Triticum aestivum L.) in Hebei province, China, and the characterization of one strain, selected for its antitumor activity. The characterization of the strain led to the identification of a novel strain of Embleya. 

My major concerns are:

-they reported two different methods to evaluate the citotoxicity. It is not clear the reason why they did not use the same method.

-they used pancreatic enzymes (line 222), it means trypsin or a mixture of enzymes?

-they report the antibacterial activity against bacteria, no bacterial types are reported.

-I would suggest to move the Table 2 to supplementary section.

-It is not clear why they do not present a table to summarize antibacterial results (line 377-382).

- I would suggest to change the title and introduction. Indeed, the authors report that they were looking for anticancer activity, while they also checked antibacterial activity, that is superficially described, but then they conclude the manuscript with the sentence "To our knowledge, this is the first report of this kind of compound with antibacterial activity against Gram-negative bacteria."

Some english mistakes are present in the manuscript, listed below:

-line 35 biological activity test were performed. singular or plural?

-line 50 Natural product antibiotics derived from various source materials. The word are seems missing.

-lines 83-87 and  lines 95-96 please rephrase.

-line 159 The atpD, recA, rpoB and trpB genes were performed. CHange with PCR of the atpD, recA, rpoB and trpB genes was performed.

-line 171 I believe that MOPS should be written in capital.

-line 319-325, please check the sentences and I was wondering why the molecules are written in capital?

-line 374, 8 (not 2) compounds  were tested!

-line 375 which the average, please check the sentence. The word "which" is frequently badly used.

-lines 425-428 please rephrase.

-line 428 Its mode faction, please correct it

Author Response

Dear reviewer,

Thank you for the valuable suggestions. In the following, we provide our itemized list of changes according to your suggestions and highlighted the changes in our manuscript.

Thank you very much for your kindness and help.

Sincerely yours,

Dr. Han Wang and Prof. Junwei Zhao

  1. they reported two different methods to evaluate the cytotoxicity. It is not clear the reason why they did not use the same method.

Reply: In the activity of the preliminary screening section, we used SRB method because of its lower cost and convenience, compared to the SRB method, CCK-8 has higher accuracy and success rate, so the cytotoxicity of the eight compounds was assayed by cell counting kit-8 (CCK-8) colorimetric method.

  1. they used pancreatic enzymes (line 222), it means trypsin or a mixture of enzymes?

Reply: Thank you very much, pancreatic enzymes were a mixture of enzymes, mainly for trypsin, pancreatic amylase and pancreatic lipase.

  1. they report the antibacterial activity against bacteria, no bacterial types are reported.

Reply: Thank you very much and we have revised the description. Please see lines 239-241.

  1. I would suggest to move the Table 2 to supplementary section.

Reply: Thank you very much and we have revised.

  1. It is not clear why they do not present a table to summarize antibacterial results (line 377-382).

Reply: Thank you very much and we have presented a table. Please see Table 4.

  1. I would suggest to change the title and introduction. Indeed, the authors report that they were looking for anticancer activity, while they also checked antibacterial activity, that is superficially described, but then they conclude the manuscript with the sentence "To our knowledge, this is the first report of this kind of compound with antibacterial activity against Gram-negative bacteria."

Reply: Thank you very much for your valuable suggestion. We have changed the title to ‘Taxonomic Characterization and Secondary Metabolite Analysis of NEAU-wh3-1: an Embleya strain with Antitumor and Antibacterial Activity’ and added antimicrobial activity in the abstract and introduction, please see title and lines36-37, 58-64, and 99-100.

As you mentioned, the purpose of our research is to obtain antitumor compound. Firstly, we isolated Streptomyces-like strains from the rhizosphere soil of wheat (Triticum aestivum L.) and then screened strains with antitumor activities; strain NEAU-wh-3-1 exhibited better inhibitory activity, so we studied its active ingredients. Consequently, one new Zincophorin analogue (compound 1) together with seven known compounds was detected. Some compounds have been reported that they have antibacterial activity. Especially Zincophorin, which structurally related to compound 1, was also referred to as M144255 or griseochellin, which possesses strong in vivo activity against Gram-positive bacteria and Clostridium coelchii. To enrich the activity spectrum of the isolated compounds, we selected three Gram-positive bacteria (Staphylococcus aureus, Bacillus subtilis and Sarcina lutea) and two Gram-negative bacteria (Klebsiella pneumoniae and Escherichia coli) to test their antibacterial activities. However, the purpose of our study also focuses on antitumor activity. So we did not add more content to part of Introduction. Thank you very much!

Some English mistakes are present in the manuscript, listed below:

  1. line 35 biological activity test were performed. singular or plural?

Reply: We have changed the presentation and please see line 35.

  1. line 50 Natural product antibiotics derived from various source materials. The word are seems missing.

Reply: We have revised and please see line 51.

  1. lines 83-87 and lines 95-96 please rephrase

Reply: We have revised and please see lines 86-92 and lines 101-102.

  1. line 159 The atpD, recA, rpoB and trpB genes were performed. CHange with PCR of the atpD, recA, rpoB and trpB genes was performed.

Reply: We have revised and please see lines 166-167.

  1. line 171 I believe that MOPS should be written in capital.

Reply: We have revised and please see line 179.

  1. line 319-325, please check the sentences and I was wondering why the molecules are written in capital?

Reply: We have checked and revised, please see lines 312-317.

  1. line 374, 8 (not 2) compounds were tested!

Reply: We have revised and please see line 364.

  1. line 375 which the average, please check the sentence. The word "which" is frequently badly used.

Reply: We have revised and please see line 365.

  1. lines 425-428 please rephrase.

Reply: We have revised and please see lines 415-418.

  1. line 428 Its mode faction, please correct it.

Reply: We have revised and please see line 418.

Reviewer 3 Report

Authors in the manuscript “Taxonomic Characterization, and Secondary Metabolite Analysis of an Embleya strain with Antitumor Activity” isolate and identify a new strain producing a new compound with strong antitumor activity against three human cell lines.

They collected rhizosphere soil samples, isolated thirty-nine strains, obtained the crude extracts of these isolates and determined their cytotoxic activity. Finally, strain NEAU-wh3-1 was selected to perform the study.

The work is interesting but it can be improved.

The title is confusing “Taxonomic Characterization, and Secondary Metabolite Analysis of an Embleya strain with Antitumor Activity”. Strain produces and compound with antitumor activity. Title should be improved.

Lines 60-61. “More than 70% of nearly 10000 known compounds are produced by Streptomyces while some rare actinobacterial genera only accounted for less than 30% [17-20]. This information is referred to bacteria or to microorganisms?

Line 78. Nouioui et al. [?].

Lines 82-87. “At present, the genus comprises only two species: Embleya scabrispora, was originally proposed as Streptomyces scabrisporus sp. nov. [41], has been reclassified to the genus Embleya as the type species [37,38], which could produce hitachimycin with antitumor, antibacterial and antiprotozoal activities [42-44]; and Embleya hyaline, first described as Streptomyces hyalinum [45,39], has been reported it could produce nybomycin which is an effective agent against antibiotic-resistant Staphylococcus aureus and is called a reverse antibiotic [46].” It should be rewritten.

Line 106. Rpm? Better production broth.

Line 165. Better Production.

Line 174. Production broth

Line 232-233. “The antibacterial activities of the isolated compounds were tested against Staphylococcus aureus, Bacillus subtilis, Klebsiella pneumoniae, Sarcina lutea and Escherichia coli with…”. Sort by gram, morphology,…..

Line 236. 3.1 Isolation and screening of an antitumor compound producing strains

Line 448. “In summary, we report a novel strain with antitumor activity,……” . Better strain producing a new compound with strong antitumor activity

Author Response

Dear reviewer,

Thank you for the valuable suggestions. In the following, we provide our itemized list of changes according to your suggestions and highlighted the changes in our manuscript.

Thank you very much for your kindness and help.

Sincerely yours,

Dr. Han Wang and Prof. Junwei Zhao

  1. The title is confusing “Taxonomic Characterization, and Secondary Metabolite Analysis of an Embleya strain with Antitumor Activity”. Strain produces and compound with antitumor activity. Title should be improved.

Reply: Thank you very much and we have changed the title to ‘Taxonomic Characterization and Secondary Metabolite Analysis of NEAU-wh3-1: an Embleya strain with Antitumor and Antibacterial Activity’.

  1. Lines 60-61. “More than 70% of nearly 10000 known compounds are produced by Streptomyces while some rare actinobacterial genera only accounted for less than 30% [17-20]. This information is referred to bacteria or to microorganisms?

Reply: Thank you very much and we have revised it in line 65, this information is referred to microorganisms

  1. Line 78. Nouioui et al. [?].

Reply: Thank you very much and we have revised. Please see line 82.

  1. Lines 82-87. “At present, the genus comprises only two species: Embleya scabrispora, was originally proposed as Streptomyces scabrisporus sp. nov. [41], has been reclassified to the genus Embleya as the type species [37,38], which could produce hitachimycin with antitumor, antibacterial and antiprotozoal activities [42-44]; and Embleya hyaline, first described as Streptomyces hyalinum [45,39], has been reported it could produce nybomycin which is an effective agent against antibiotic-resistant Staphylococcus aureus and is called a reverse antibiotic [46].” It should be rewritten.

Reply: Thank you very much and we have revised. Please see lines 86-92.

  1. Line 106. Rpm? Better production broth.

Reply: Thank you very much and we have revised. Please see line 111-114.

  1. Line 165. Better Production.

Reply: Thank you very much and we have revised. Please see line 173.

  1. Line 174. Production broth

Reply: Thank you very much and we have revised. Please see line 182.

  1. Line 232-233. “The antibacterial activities of the isolated compounds were tested against Staphylococcus aureus, Bacillus subtilis, Klebsiella pneumoniae, Sarcina lutea and Escherichia coli with…”. Sort by gram, morphology,…..

Reply: Thank you very much and we have revised. Please see line 239-241.

  1. Line 236. 3.1 Isolation and screening of an antitumor compound producing strains

Reply: Thank you very much and we have revised. Please see line 244.

  1. Line 448. “In summary, we report a novel strain with antitumor activity,……” . Better strain producing a new compound with strong antitumor activity

Reply: Thank you very much and we have revised. Please see line 439.

Round 2

Reviewer 2 Report

Dear Authors,

please check the sentence in line 319 "which structures were elucidated".

Line 243-245: which strains did you use? Reference type, ATCC? Or clinically or environmental isolated strains? You should mention this.

The authors improved the manuscript following all the suggestions.

thanks